# Cell-Based Measurement of Mitochondrial Function in Human Airway Smooth Muscle Cells

**DOI:** 10.3390/ijms241411506

**Published:** 2023-07-15

**Authors:** Sanjana Mahadev Bhat, Jane Q. Yap, Oscar A. Ramirez-Ramirez, Philippe Delmotte, Gary C. Sieck

**Affiliations:** Department of Physiology & Biomedical Engineering, Mayo Clinic, Rochester, MN 55905, USA; mahadevbhat.sanjana@mayo.edu (S.M.B.); yap.jane@mayo.edu (J.Q.Y.); oramirezramirez@uwhealth.org (O.A.R.-R.); delmotte.philippe@mayo.edu (P.D.)

**Keywords:** airway smooth muscle, succinate dehydrogenase, mitochondria, oxygen consumption, confocal microscope

## Abstract

Cellular mitochondrial function can be assessed using high-resolution respirometry that measures the O_2_ consumption rate (OCR) across a number of cells. However, a direct measurement of cellular mitochondrial function provides valuable information and physiological insight. In the present study, we used a quantitative histochemical technique to measure the activity of succinate dehydrogenase (SDH), a key enzyme located in the inner mitochondrial membrane, which participates in both the tricarboxylic acid (TCA) cycle and electron transport chain (ETC) as Complex II. In this study, we determine the maximum velocity of the SDH reaction (SDH_max_) in individual human airway smooth muscle (hASM) cells. To measure SDH_max_, hASM cells were exposed to a solution containing 80 mM succinate and 1.5 mM nitroblue tetrazolium (NBT, reaction indicator). As the reaction proceeded, the change in optical density (OD) due to the reduction of NBT to its diformazan (peak absorbance wavelength of 570 nm) was measured using a confocal microscope with the pathlength for light absorbance tightly controlled. SDH_max_ was determined during the linear period of the SDH reaction and expressed as mmol fumarate/liter of cell/min. We determine that this technique is rigorous and reproducible, and reliable for the measurement of mitochondrial function in individual cells.

## 1. Introduction

Succinate dehydrogenase (SDH) is a key enzyme located in the inner mitochondrial membrane and participates in both the tricarboxylic acid (TCA) cycle and the electron transport chain (ETC) as Complex II. SDH mediates the oxidation of succinate to fumarate in the TCA cycle, which is coupled to the reduction of ubiquinone to ubiquinol in the ETC [1]. Previously, we developed a quantitative histochemical technique for determining the maximum velocity of the SDH reaction (SDH_max_) in single skeletal muscle fibers [2,3,4,5]. In this quantitative histochemical technique, the transport of electrons to O_2_ (cytochrome oxidase/complex IV) is inhibited using azide, and an exogenous electron carrier, 1-methoxyphenazine methosulphate (mPMS), is provided to facilitate the specific reduction of the reaction indicator nitroblue tetrazolium (NBT) to its diformazan (NBT_dfz_) [2,3]. The progressive accumulation of NBT_dfz_ is then measured repeatedly by the absorbance of light (at 570 nm—peak absorbance for NBT_dfz_) (Figure 1). Accordingly, as the SDH reaction proceeds, the optical density (OD) within the cell increases with the accumulation of NBT_dfz_. By optimizing the chemical reaction (e.g., unlimited substrate), the change in OD with time reflects SDH_max_, which can be determined for a cell or a region within the cell, using the Beer–Lambert equation in which the pathlength for light absorbance is critical.

Previously, SDH_max_ was performed in transverse sections of skeletal muscle fibers cut at 6 µm thickness by cryo-sectioning, thereby controlling pathlength for light absorbance [2,3,6,7,8,9]. However, in live intact cells cultured in vitro, it is more difficult to control the pathlength for light absorbance. In the present study, we use confocal microscopy to optically section the cell, which allows control of the pathlength for light absorbance within the cell. Confocal microscopy is a convenient tool for obtaining a series of two-dimensional (2D) images (optical slices) that can be reconstructed in three dimensions (3D). Therefore, confocal optical sectioning provides an opportunity for volumetric measurements within the cell, e.g., mitochondrial volume [10].

Previously, we found that SDH_max_ varies across skeletal muscle fiber types and changes with activity, age, and disease conditions [2,3,5,6,8,9,11,12,13,14,15,16,17,18,19,20,21,22]. Differences in SDH_max_ across skeletal muscle fibers corresponded to differences in mitochondrial volume density [6,8,9]. Such differences in mitochondrial volume density are important to consider as they impact mitochondrial function. Recently we found that exposing human airway smooth muscle (hASM) cells to TNFα increases mitochondrial volume density disproportionately to the change in oxygen consumption rate (OCR) [23,24]. Thus, when OCR was normalized for TNFα-induced changes in mitochondrial volume density, the respiratory capacity of hASM cells was reduced. In the present study, we assess the relationship between SDH_max_ and mitochondrial volume density in individual hASM cells, which has not been previously explored.

The goal of the present study was to validate a novel microscopic optical sectioning technique to measure SDH_max_ in individual hASM cells based on the progressive precipitation of NBT_dfz_. In the present study, the relationship between SDH_max_ and mitochondrial volume density was examined. In addition, measurements of SDH_max_ were compared to respirometry-based OCR measurements. Finally, the rigor and reproducibility of SDH_max_ measurements across hASM cells were assessed both within and across patients. 

## 2. Results

### 2.1. Measurement of SDH Reaction and Substrate Dependence in hASM Cells

The substrate dependency of the SDH reaction was assessed in both saponin-permeabilized and intact hASM cells with succinate concentrations ranging from 0 mM to 120 mM. The SDH reaction was assessed by measuring changes in OD within delineated hASM cells every 15 s across a 10-min period. In both permeabilized and intact hASM cells, the reaction progressed linearly from 0 to 8 min with the progressive accumulation of NBT_dfz_ (change in OD) with time (R^2^ = 0.99; Figure 2A,B). In addition, in both cell preparations, we measured the change in OD in the absence of succinate (without substrate). We observed that the reaction without substrate was also linear with respect to the time of incubation (R^2^ = 0.98) and was significantly lower compared to the change in OD in the presence of succinate (*p* < 0.0001; Figure 2B). The change in OD in the absence of cells (i.e., buffer blanks) was also measured and was found to be unchanged throughout the incubation period. In saponin-permeabilized hASM cells, SDH_max_ was achieved at a succinate concentration of 10 mM whereas in intact hASM cells, SDH_max_ was achieved at a succinate concentration of 80 mM. When SDH_max_ values were compared between intact and permeabilized hASM cells, the SDH_max_ values from the same patient were comparable albeit at different succinate concentrations (Figure 2C). Importantly, these results validate that SDH_max_ can be measured in intact hASM cells.

### 2.2. Assessment of the Distribution of NBT_dfz_ Precipitate within hASM Cells

The distribution of NBT_dfz_ precipitate within intact hASM cells was estimated by performing optical slicing (pathlength for light absorbance in Equation (2)) to obtain a 3D image of the NBT_dfz_ precipitate distributed throughout the hASM cell. The sampling depth was restricted to 5 µm in the z-axis with a fixed optical slice thickness of 0.5 µm (Figure 3A). Each Z-series of optical slices was obtained at 15 s intervals over a 10 min period. In one series of studies, a step size of 0.5 µm was used, and the total acquisition time for 10 optical slices in the Z-stack through 5 µm of cell thickness was ~11 s (Figure 3B). In another series of studies, the optical slice was kept at 0.5 µm but the step size was increased to 1.0 µm, thus reducing the number of optical slices obtained (Figure 3C). By reducing the number of optical slices, the total acquisition time was decreased to ~3 s. Importantly, the time interval between imaging each optical slice in the Z-series was the same (15 s). The precipitation of NBT_dfz_ measured as OD across all optical slices was linear and did not significantly differ across time using both 0.5 µm and 1.0 µm step sizes (Figure 3D). Using either step size, the average SDH_max_ across optical slices of the Z-stack was comparable (Figure 3E).

### 2.3. OD Measured across Time Is Proportional to Pathlength of Light Absorbance

The relationship between pathlength and OD was determined in intact hASM cells by performing optical slicing to obtain a 3D image of the NBT_dfz_ precipitate distributed throughout the hASM cell with varying optical slice thickness. The sampling depth was restricted to 5 µm in the z-axis with a fixed step size of 0.5 µm between optical slices (Figure 4A). The aperture settings were adjusted to provide an optical slice thickness of 0.5 µm or 1.0 µm (Figure 4B,C). Each optical slice of the Z-series was obtained at every 15 s interval over a 10 min period and the time interval between imaging each optical slice in the Z-series was the same (15 s). The OD across all optical slices was linear and did not significantly differ across time in both 0.5 µm and 1.0 µm optical slice thickness (Figure 4D). In both conditions of optical slice thickness, the average SDH_max_ across optical slices of the Z-stack was comparable (Figure 4E). 

### 2.4. SDH_max_ Measurements Are Reproducible across Patients

SDH_max_ measurements were performed in 30 intact hASM cells from each patient to test the reproducibility of the assay. The variability of SDH_max_ measurements in cells within a single patient was relatively low, with the coefficient of variation (CV) averaging ~6% across intact hASM cells and ~4% across permeabilized cells (Figure 5, Table 1).

### 2.5. Mitochondrial Volume Density Is Variable across hASM Cells

To simultaneously measure SDH_max_ and mitochondrial volume density, intact hASM cells were loaded with MitoTracker^®^ Green FM and SDH_max_ was measured within the same hASM cell (Figure 6A,B). SDH_max_ measured in hASM cells with MitoTracker labeled mitochondria showed decreased SDH_max_ when compared to unlabeled hASM cells. To avoid the decrease in SDH_max_, in a separate set of experiments, the mitochondrial volume density in intact hASM cells was measured from independent determinations of mitochondrial volume (Figure 6C). The mitochondrial volume density across 30 hASM cells per patient showed significant variation between patients with an average CV of ~14% (Figure 6C, Table 2). The relationship between SDH_max_ and mitochondrial volume density was analyzed by Pearson’s correlation (Figure 6D). Across patients, there was no linear relationship between mean SDH_max_ and mean mitochondrial volume density (slope = 0.8, R^2^ = 0.62, *p* > 0.05).

### 2.6. Mitochondrial Volume Density within Individual hASM Cells Affects SDH_max_


To circumvent the effect of the MitoTracker on SDH_max_ measurements, we tested the use of CellLight™ Mitochondria-GFP for the simultaneous measurement of SDH_max_ in individual hASM cells. To validate that the pattern of CellLight, labeling was restricted to mitochondria; hASM cells were loaded with both the CellLight™ Mitochondria-GFP and MitoTracker^®^ Red FM (Figure 7A,B). Mitochondrial labeling with CellLight and MitoTracker showed a similar pattern of labeling (Figure 7C). However, when mitochondrial volume density was measured in the hASM cells that showed homogenous labeling, CellLight labeling estimated a higher mitochondrial volume density when compared to MitoTracker labeling (*p* < 0.0001; Figure 7D). When the relationship between mitochondrial volume density measured with CellLight and MitoTracker labeling was analyzed by Pearson’s correlation, we observed a significant positive linear relationship across cells (slope = 0.98, R^2^ = 0.95, *p* < 0.0001; Figure 7E).

In separate experiments, SDH_max_ was measured in intact hASM cells with CellLight labeled mitochondria. SDH_max_ measurements in CellLight labeled hASM cells were comparable to unlabeled hASM cells (Figure 7F). The relationship between SDH_max_ and mitochondrial volume density measured in the same cell was analyzed by Pearson’s correlation (Figure 7G). Across cells, there was a significant positive linear relationship between SDH_max_ and mitochondrial volume density measured (slope = 0.93, R^2^ = 0.87, *p* < 0.0001). 

### 2.7. SDH_max_ Compares with OCR Measurements When Normalized to Mitochondrial Volume

The raw OCR measurements were obtained from intact hASM cells from the same patients (Figure 8A). In the present study, the number of adherent cells per well was estimated both before and after the mitochondrial stress test. Overall, the adherent cell count was reduced by ~50% when compared between pre- and post-stress test counts. This change in adherent cell count also affected the normalized OCR measurements, where the OCR normalized to pre-stress test cell counts were ~2-fold lower compared to OCR normalized to post-stress test cell counts (Figure 8B, Table 3). Of interest, the qualitative changes in OCR during the stress test were similar whether normalized for pre- or post-stress test cell counts. In addition, the OCR measurements were found to be more variable both within and across patients. The CV of the OCR measurements per cell averaged at ~11% across patients compared to ~6% for SDH_max_ in intact hASM cells (Table 3). The relationship between SDH_max_ and OCR per cell was analyzed by Pearson’s correlation (Figure 8C). Across patients, there was no significant linear relationship between mean SDH_max_ and mean OCR measurements per cell (slope = 0.32, R^2^ = 0.10, *p* > 0.05). In addition, the maximum OCR measurements obtained were normalized to mean mitochondrial volume density (MVD), measured from independent determinations of mitochondrial volume (Figure 8D, Table 3). When the relationship between SDH_max_ and OCR per mitochondrion was analyzed by Pearson’s correlation, we observed a significant positive linear relationship across patients (slope = 0.98, R^2^ = 0.96, *p* < 0.001; Figure 8E).

## 3. Discussion

In the present study, we validate the use of a quantitative histochemical technique to measure the maximum velocity of the SDH reaction (SDH_max_) in individual hASM cells. The method used in this study was similar to that previously used to measure SDH_max_ in skeletal muscle fibers [2,3,4,5,6,8,9,11,12,13,14,15,16,17,18,19,20,21,22,25]. However, in addition to the measurements being made in different tissues, the novelty of measuring SDH_max_ in hASM cells involved the control of pathlength for light absorbance by confocal-based optical slicing rather than cryo-sectioning (e.g., in skeletal muscle fibers), which is critical in the quantification of the SDH reaction rate. The main findings of the present study were: (1) the SDH reaction rates in both intact and saponin-permeabilized hASM cells were linear across an 8 min period; (2) the optimal succinate concentration for measuring SDH_max_ was ~10 mM in permeabilized hASM cells compared to ~80 mM for intact hASM cells; (3) at optimal succinate concentrations, the SDH_max_ values for intact and permeabilized hASM cells from the same patient were comparable; (4) SDH_max_ varies with mitochondrial volume density across hASM cells; and (5) when SDH_max_ was compared to OCR measurements normalized for mitochondrial volume density, there was a strong correlation (R^2^ = 0.96). We conclude that measurements of SDH_max_ provide a reliable estimate of the maximum respiratory capacity of individual hASM cells.

### 3.1. Imaging-Based Measurements of SDH_max_ in hASM Cells

The technique for measuring SDH_max_ was first developed and reported in 1986 by Sieck et al. [2], followed by a subsequent publication that provided greater detail [3]. The quantitative histochemical technique measures the progressive reduction of NBT (reaction indicator) to its diformazan state (NBT_dfz_). The reaction is performed using mPMS as an exogenous electron carrier and azide to inhibit cytochrome oxidase (Complex IV), thereby reducing the non-specific reduction of NBT. The optimal concentration of the various compounds used in the assay has been tested in previous studies and has been consistently shown to give reliable results in various testing models [2,3,4,5,6,8,9,11,12,13,14,15,16,17,18,19,20,21,22,25]. The present study extends the use of the assay in in vitro cultures by using primary hASM cells. Importantly, in individual cells, the results of the present study demonstrate that confocal optical slicing can be used to control the pathlength for light absorbance, which is critical in the Beer–Lambert equation.

In the quantitative histochemical method for measuring SDH_max_, the Beer–Lambert equation requires control of the pathlength for light absorbance in determining the accumulation of the reaction product (i.e., NBT_dfz_). In measuring SDH_max_ in skeletal muscle fibers, the pathlength for light absorbance was controlled by cryo-sectioning the fibers at a defined thickness [6,8,9,22]. Cryo-sectioning is difficult if not impossible in cultured cells such as hASM cells; thus, in the novel technique we validated, a confocal microscope, which was used to acquire optical slices of a defined depth, thereby controlling pathlength for light absorbance. Furthermore, by acquiring a Z-series of optical slices through a cell, we obtained information regarding the volumetric distribution of the reaction product. Three-dimensional confocal imaging techniques have been previously reported in detail [10,26,27]. However, one major caveat in obtaining repetitive 3D images is the time consumed for image acquisition. To circumvent this limitation, we demonstrated that fewer optical slices can be obtained to determine the volumetric accumulation and distribution of the reaction product, thus reducing the time of light exposure by half. Importantly, we observed similar SDH_max_ measurements regardless of the number of optical slices sampled or the thickness of the optical slices. The similar measurements across optical slices indicate that NBT_dfz_ is distributed within the cytosol rather than within the mitochondria and the change in OD is proportional to the thickness of the optical slices.

In the quantitative histochemical technique, it is essential to use an optimal concentration of succinate to avoid substrate limiting the reaction. In intact hASM cells, we found that SDH_max_ was achieved at a succinate concentration of 80 mM in the reaction media. However, the exogenous succinate concentration in intact hASM cells does not necessarily reflect the available succinate for the SDH reaction. Therefore, we compared the optimal succinate concentration for SDH_max_ in intact hASM cells to that in saponin-permeabilized cells, where the cell membrane diffusion barrier was removed. Since saponin can also permeabilize the mitochondrial membrane, we carefully titrated the concentration of saponin and confirmed that the mitochondrial membrane potential was not affected. In permeabilized hASM cells, SDH_max_ was achieved at a succinate concentration of 10 mM. Importantly, at optimal succinate concentrations, the SDH_max_ values measured in intact and permeabilized hASM cells were comparable. This indicated that succinate transport across the cell membrane is necessary, and most likely achieved by a Na^+^-dependent dicarboxylic acid transporter [28,29,30,31]. The comparable SDH_max_ measurements further indicated that an adequate amount of succinate is being transported into intact cells to achieve SDH_max_. Furthermore, in both intact and permeabilized hASM cells, we confirmed that the SDH reaction is linear over an 8 min period. The temporal linearity of the SDH reaction is similar to that found in skeletal muscle fibers [2,3,6,9,22]. Thus, at maximum substrate concentration, the reaction proceeds in a linear fashion from which the maximum velocity of SDH can be determined. Additionally, in the present study, we assessed the reproducibility of SDH_max_ in hASM cells in each patient. Based on the sample size used, we observed that the variability of SDH_max_ measurements across hASM cells within a patient was less than that observed across patients.

### 3.2. SDH_max_ Changes with Mitochondrial Volume Density in hASM Cells

In the present study, we used 3D confocal imaging to identify mitochondria and measure mitochondrial volume density within hASM cells across patients. We observed significant variability in mitochondrial volume density across hASM cells from a single patient implying the presence of inherent differences in mitochondrial volume between individual hASM cells and patients. In previous studies, we compared the use of 3D confocal imaging to determine mitochondrial volume to that obtained by electron microscopy (EM) and found the mitochondrial volume densities determined by the two methods to be comparable [7]. However, MitoTracker labeling and 3D imaging have a major advantage over EM in that many cells can be analyzed per sample, and MitoTracker allows measurements from living cells [7,9,23,32]. 

In previous studies, we observed that mitochondrial volume density is greater in hASM during various inflammatory conditions, which reflects an increase in energy demand within hASM cells [23,24,33,34,35]. In skeletal muscle, we previously reported differences in mitochondrial volume densities between type I and IIa fibers vs. type IIx/IIb fibers consistent with their use and activation history [9]. Type I and IIa diaphragm muscle fibers are frequently active and have higher mitochondrial volume densities and SDH_max_ values compared to less active type IIx/IIb fibers. Imposing reduced activity on diaphragm muscle fibers reduces mitochondrial volume density and SDH_max_ of type I and IIa fibers (15, 17, 18, 31). These previous results indicate that mitochondria are highly dynamic organelles undergoing cycles of fission/fragmentation and fusion as well as mitochondrial biogenesis to meet energy demands [23,36,37,38,39].

Normalizing SDH_max_ to the mitochondrial volume provides important information on the relationship between mitochondrial structure and function. SDH_max_ normalized for mitochondrial volume density is much higher in type I and IIa diaphragm muscle fibers compared to type IIx/IIb fibers, thus reflecting a higher maximum respiratory capacity per mitochondrion in these fibers (18). Unfortunately, simultaneous measurements of SDH_max_ and mitochondrial imaging by MitoTracker labeling are not possible in hASM cells since we found that MitoTracker decreased SDH_max_. Thus, MitoTracker labeling appears to impact mitochondrial function, perhaps because mitochondrial labeling depends initially on the proton gradient followed by binding to thiol groups in the mitochondrial matrix. By comparison, CellLight, a BacMam expression vector encoding GFP fused to the leader sequence of E1α pyruvate dehydrogenase [40], labels independently of the mitochondrial membrane potential, unlike MitoTracker [41,42]. When comparing results obtained from MitoTracker versus CellLight labeling in hASM cells from the same patient, we observed that the mitochondrial volume density measured with CellLight was greater compared to that measured with MitoTracker. However, unlike MitoTracker, we observed no effect of CellLight labeling on SDH_max_ measurements. Additionally, the relationship between SDH_max_ and mitochondrial volume density determined by CellLight labeling was much stronger than that observed using MitoTracker. 

### 3.3. SDH_max_ Measurements Correlate to Normalized OCR Measurements

In the present study, maximum OCR measurements normalized to cell count did not correlate to the average SDH_max_ within patients. However, when OCR measurements were normalized to mitochondrial volume density there was a strong correlation to the average SDH_max_ across cells. Studies have also provided evidence that NBT_dfz_ is specific for measuring SDH activity and thus effective for determining mitochondrial oxidative activity [28,30,43,44,45,46,47]. Additionally, the underlying principles of this cell-based measurement of SDH_max_ are comparable to those in lysate-based biochemical measurements [3,9,48,49,50,51]. Thus, the use of the quantitative histochemical technique to measure SDH_max_ provides measurements of mitochondrial respiratory capacity within individual cells with lower variability.

### 3.4. Limitations of SDH_max_ Measurements

While our results confirm that SDH_max_ measurement is a reliable technique, we acknowledge the presence of certain limitations associated with the assay. We show evidence that the measurement of SDH_max_ reflects the maximum respiratory capacity of mitochondria in hASM cells but does not imply that SDH_max_ is equivalent to OCR measurements. Unlike respirometry-based OCR measurements, SDH_max_ does not inform us how much of the respiratory capacity measured is used to meet the cellular energy demands, ATP-linked respiration, and proton leak, among others.

## 4. Material and Methods

### 4.1. Ethics Statement

The study was reviewed by Institutional Review Board at Mayo Clinic (IRB #16-009655) and considered to be exempt for the following reasons. Patient consent was obtained during pre-surgical evaluation (in the Pulmonary, Oncology, or Thoracic Surgery Clinics) prior to surgery in non-threatening environments and patients remained anonymous. No one on the team was involved in patient selection or in the surgery itself. In no cases were patients contacted by any of the investigators involved in this proposal. Patient history was provided at the time of tissue acquisition for the recording of sex, demographics, pulmonary disease status, pulmonary function testing, imaging, co-morbidities, and medications, but no patient identifiers were stored. Accordingly, samples were numbered, but patient identifiers were not attached to the samples, preventing retrospective patient identification from samples alone.

### 4.2. Dissociation of Cells from Bronchiolar Tissue 

Bronchial tissue samples were obtained from anonymized female and male patients undergoing lung resection surgeries. The selected patients had no history of asthma or other chronic lung diseases. After pathological assessment, normal regions were identified, and third to sixth-generation bronchi were isolated. The smooth muscle layer was dissected under a microscope, and hASM cells were dissociated by enzymatic digestion using papain and collagenase with ovomucoid/albumin separation as per the manufacturer’s instructions (Worthington Biochemical, Lakewood, NJ, USA) as previously described [52,53]. hASM cells were maintained at 37 °C (5% CO_2_—95% air) in phenol red-free DMEM/F-12 medium (Invitrogen, Carlsbad, CA, USA) supplemented with 10% fetal bovine serum (FBS; Invitrogen, Carlsbad, CA, USA) and 100 U/mL of penicillin/streptomycin (Invitrogen, Carlsbad, CA, USA) in sterile culture dishes. Isolated hASM cells from 1 to 3 subcultures were used in all experiments. Before experiments, the cells were serum deprived for 48 h by changing the growth medium to DMEM/F-12 medium lacking FBS.

### 4.3. Confirming hASM Phenotype by Immunoreactivity to α-Smooth Muscle Actin (α-SMA)

Before conducting any experiments, the phenotype of hASM cells was confirmed by immunocytochemical analysis of α-smooth muscle actin (α-SMA) expression as previously described [34,54,55,56]. The number of cells expressing α-SMA compared to the total number of 4′,6-diamidino-2-phenylindole, dihydrochloride (DAPI) labeled nuclei within cells were identified for each patient. In the histochemical analysis, cells were plated at a density of ~10,000 cells/well in a Nunc™ Lab-Tek™ II Chamber 8-well multi-chamber slide (Thermo Fisher Scientific, Rockford, IL, USA), incubated until cell adherence was achieved and serum deprived. Cells were fixed with 4% paraformaldehyde in 1X phosphate-buffered saline (PBS) for 10 min at room temperature and washed with 1X PBS. Cells were then blocked for 1 h using a blocking buffer containing 10% normal donkey serum (Sigma-Aldrich, St. Louis, MO, USA), 0.2% triton X-100, and 1X PBS. The cells were then incubated overnight at 4 °C with anti-αSMA (1:500 dilution, rabbit polyclonal) (Abcam, Boston, MA, USA) antibody in a diluent solution (2.5% normal donkey serum, 0.25% sodium azide, 0.2% triton X-100, 1X PBS). Following this, the cells were incubated with donkey anti-rabbit biotin-conjugated secondary antibody (1:400, diluted in antibody diluent) (Jackson Immunoresearch, West Grove, PA, USA) for 1 h at room temperature, followed by Streptavidin Alexa Fluor 568 (1:200 in PBS; Invitrogen, Carlsbad, CA, USA). The cells were mounted using Fluoro-Gel II mounting medium with DAPI (Electron Microscopy Sciences, Hatfield, PA, USA) and imaged using a Nikon Eclipse Ti inverted microscope (×60/1.4 NA oil-immersion objective).

Dissociated cells were imaged to distinguish α-SMA expressing hASM cells and the percentage of hASM cells was determined as a fraction of total dissociated cells (determined from DAPI) (Figure 9). Consistent with our previous studies [34,54,55,56], the vast majority (90–95%) of dissociated cells expressed α-SMA, and these hASM cells were also larger and displayed a characteristic elongated shape (Figure 9A). Patient samples that did not contain >90% confirmed hASM cells were excluded from further analyses (three out of nine patient samples) (Figure 9B). In additional experiments involving mitochondrial imaging and quantitative histochemical analysis, the distinct morphological features were used to identify hASM cells. 

### 4.4. Quantitative Histochemical Measurement of SDH Reaction Velocity 

For the quantitative histochemical procedure, the Nikon Eclipse A1 laser scanning confocal microscope with a ×60/1.4 NA oil-immersion objective at 12-bit resolution into a 1024 × 1024-pixel array using the transmitted light channel was used for image acquisition. An interference filter with a peak emission wavelength of 570 nm was placed in the light path to limit the spectral range of the light source to the maximum absorption wavelength of NBT_dfz_ [57]. 

The measured gray level (GL) of the microscope was calibrated to known OD units using a photographic density stepwedge tablet (Figure 10) (0.04–2.20 OD units in increments of 0.15 OD; Stouffer Industries, Mishawaka, IN, USA). The dynamic range of the microscope was adjusted to take advantage of the full range of OD while avoiding saturation of the images at both ends of the OD range. This was performed by adjusting the gain of the photomultiplier while maintaining the intensity of the LED light source and setting the black level to 2.20 OD. The photomultiplier gain (brightness level) was adjusted such that the GL at 0.04 OD was not saturated. The GL within regions of interest (ROI) of the stepwedge tablet was then mapped to OD using the NIS-Elements Analysis software (Version 5.20.02) (Nikon Instruments Inc., Melville, NY, USA). Figure 10 shows the relationship between known (stepwedge) OD values and measured GL. The data were analyzed by a four-parameter general curve fit function to derive an equation (Equation (1)) that was then used to transform measured GL values to OD.
(1)$$OD=\frac{8.22}{(1.12\times 10^{-7})+GL^{0.09}}$$

Following calibration, the quantitative histochemical procedure for measuring the maximum velocity of the SDH reaction was performed in hASM cells. This technique has been previously described in detail in single skeletal muscle fibers [2,3,4,5,11,12,13,14,15,16,17,19,20,21,25]. hASM cells were plated in 8-well Ibidi μ-slide plates (Ibidi GmbH, Gewerbehof Gräfelfing, Germany) at a density of ~15,000 cells/well and incubated to allow for cell adherence, followed by serum deprivation. In the quantitative histochemical procedure, the concentration of succinate must be sufficient to avoid substrate-limiting the SDH reaction. In intact hASM cells, the plasma membrane presents a barrier for substrate diffusion. To determine the adequacy of succinate transport in hASM cells, the SDH_max_ in intact cells was compared to that of saponin-permeabilized cells (see below for details).

Intact and permeabilized hASM cells from the same patients were exposed to solutions containing either no succinate (without substrate) or a succinate-containing solution. Both solutions contained (in mM): 1.5 NBT (reaction indicator), 5 Ethylenediaminetetraacetic acid (EDTA), 1 mPMS, and 0.75 azide in 0.1 M phosphate buffer (pH: 7.4). In the quantitative histochemical procedure, the progressive precipitation of NBT_dfz_ due to the reduction of NBT is used as the reaction indicator. Experiments were performed at room temperature (22 ± 1 °C), with similar parameters maintained across preparations. hASM cells were delineated as the ROI while the nucleus was excluded, and NBT_dfz_ precipitation was measured as the progressive change in OD as the SDH reaction proceeded. The change in average OD within the selected ROI was measured every 15 s across a 10 min period. The SDH_max_ was then determined using the Beer–Lambert equation (below):(2)dNBTdfzfumaratedt=dODdtkl
where OD is measured based on the calibrated gray level of NBT_dfz_ accumulation, k is the molar extinction coefficient for NBT_dfz_ (26,478 M^−1^ cm^−1^), and l is the pathlength for light absorbance (0.5 or 1.0 µm optical slices). The SDH_max_ is expressed as millimoles of fumarate per liter of cell per min.

The maximum substrate (succinate) concentration required for measuring SDH_max_ was previously determined in skeletal muscle fibers [2,3] but not in hASM cells. In a separate series of experiments, the maximum succinate concentration required to determine SDH_max_ in intact and saponin-permeabilized hASM cells was compared. Prior to performing the SDH assay, we assessed saponin-induced permeabilization of the cell membrane by the uptake of trypan blue in hASM cells. We also assessed any potential disruption of the mitochondrial membrane by using TMRM to monitor the mitochondrial membrane potential during saponin titration. For permeabilization of cells, isolated hASM cells were first incubated in serum-free medium containing 25 µg/mL of saponin for ~1 min at room temperature to allow permeabilization. The media was aspirated, and permeabilized hASM cells were then incubated with different succinate concentrations (0 mM, 10 mM, 20 mM, 40 mM, 80 mM, and 120 mM), and the rate of the SDH reaction was measured as described above. For comparison, intact hASM cells from the same patient were incubated with the same succinate concentrations (0 mM, 10 mM, 20 mM, 40 mM, 80 mM, and 120 mM) and the rate of the SDH reaction was measured as described above.

To reduce time of acquisition and assess the impact of 3D confocal sampling parameters on SDH_max_ measurements, two sampling parameters were used to obtain 3D confocal images during the SDH assay. First, a step size of 0.5 µm or 1.0 µm between optical slices were obtained while controlling the aperture setting to provide a 0.5 µm optical slice thickness (Figure 3A–C). Second, the z-axis depth of an optical slice was controlled by adjusting the aperture setting to provide a 0.5 µm or 1.0 µm optical slice thickness while controlling the step size to 0.5 µm between each optical slice (Figure 4A–C). In both conditions, the ROIs within the hASM cells were sampled along the z-axis with the total sampling depth limited to 5 µm. Previously, we examined the impact of increasing step size between optical slices on volume measurements using confocal microscopy and observed ~10% z-axis distortion [26,27]. In both sampling parameters, for each cell, the “starting” and “ending” optical slices of the Z-stack were determined from where the nucleus of the cell was clearly visible (mid-nuclear optical slice). The OD and the SDH_max_ measured in each optical slice (the pathlength for light absorbance) were measured within each cell as described above. 

In all assays, multiple hASM cells (identified by size and shape) were visualized within a single microscopic field. Due to the large size of hASM cells, the borders of some cells overlapped, and we analyzed only those hASM cells whose borders were not overlapping. Typically, this selection process resulted in analyzing 3 to 4 hASM cells per field. Based on an a priori power analysis of variance in SDH_max_ measurements in hASM cells, 20–30 hASM cells were analyzed in each of the 6 bronchial samples (patients). 

### 4.5. Labeling of Mitochondria with MitoTracker in hASM Cells

hASM cells were plated at a density of ~15,000 cells/well into 8-well Ibidi μ-slide plates (Ibidi GmbH, Gewerbehof Gräfelfing, Germany) and incubated to allow for cell adherence and subsequently serum deprived. The cells were loaded with 200 nM MitoTracker^®^ Red FM (Invitrogen, Carlsbad, CA, USA; excitation/emission wavelength: 581/644 nm) or 200 nM MitoTracker^®^ Green FM (Invitrogen, Carlsbad, CA, USA; excitation/emission wavelength: 490/516 nm) and 1 µg/mL Hoechst 33342 Solution (Invitrogen, Carlsbad, CA, USA; excitation/emission wavelength: 361/486 nm), to label the nucleus, in serum-free media pre-warmed to 37 °C, for 15 min followed by extensive washing with 1X Hanks’ Balanced Salt Solution (HBSS). 

### 4.6. Transduction-Mediated Labeling of Mitochondria in hASM Cells

hASM cells were plated at a density of ~15,000 cells/well into 8-well Ibidi μ-slide plates (Ibidi GmbH, Gewerbehof Gräfelfing, Germany) and incubated to allow for cell adherence. Adhered hASM cells were transduced with CellLight™ Mitochondria-GFP, BacMam 2.0 (Invitrogen, Carlsbad, CA, USA) to label the mitochondria as per the manufacturer’s instructions. This is a baculovirus system with a fusion construct of the leader sequence of E1 alpha pyruvate dehydrogenase and emGFP, providing accurate and specific targeting to cellular mitochondria. hASM cells were incubated in growth media containing the CellLight reagent for 24 h at 37 °C. Subsequently, the hASM cells were washed with 1X PBS and serum deprived for 48 h for further experimentation. Although the efficiency of transduction was relatively low (~60%), GFP protein could be observed in the mitochondria.

### 4.7. Confocal Imaging of Mitochondria in hASM Cells

Mitochondria in hASM cells (distinguished by their size and shape) were imaged using the same confocal microscope system used for SDH measurements as described above [7,23,32,34,58]. The dynamic range for imaging was set by first scanning a region containing no fluorescence signal and then a second region of interest containing maximum fluorescence. A series of 0.5 µm optical slices were acquired for each hASM cell. The Z-stack was acquired and analyzed using NIS-Elements software (Version 5.20.02) (Nikon Instruments Inc., Melville, NY, USA). Multiple hASM cells were visualized in each microscopic field. Based on an a priori power analysis of variance in mitochondrial volume density measurements in hASM cells, 20–30 hASM cells were analyzed in each of the 6 bronchial samples (patients). 

### 4.8. Mitochondrial Volume Density 

The 3D images obtained were deconvolved by using the automatic deconvolution algorithm on NIS-Elements analysis software (Version 5.20.02) (Modified Richardson Lucy method; Point Scan Confocal modality; Nikon Instruments Inc., Melville, NY, USA) [7,23,34,59]. Deconvolution is known to improve the signal-to-noise ratio in the images and thereby improve contrast and edge detection. The voxel dimension of each deconvolved optical slice was 0.207 × 0.207 × 0.5 µm. After deconvolution, the boundaries of each hASM cell were delineated in ImageJ-Fiji software (Version 1.53t) (https://imagej.nih.gov/ij/). Using the mitochondrial analyzer plugin on ImageJ, the Z-stacks were then processed for background correction and ridge filter detection [23,32,60,61,62]. The mitochondria within hASM cells were identified by thresholding to create a binary image and then skeletonized for morphometric analysis. Using the thresholded image, total mitochondrial volume was measured by the mitochondrial analyzer plugin, where the number of voxels containing fluorescently labeled mitochondria within a single delineated hASM cell was determined [32,63]. Mitochondrial volume density was calculated as the ratio of mitochondrial volume within the cell to the total volume of the delineated hASM cell [6,7,8,9,23,59].

### 4.9. Measurement of Oxygen Consumption Rate (OCR) by Respirometry

hASM cells were plated into the Seahorse XF24 cell culture microplates (Agilent Technologies, Santa Clara, CA, USA) as recommended by the manufacturer for optimal adherence and growth of mammalian cells. hASM cells were then incubated to allow for cell adherence and formation of a confluent monolayer, and subsequently, serum-deprived for 48 h. Following serum deprivation, oxygen consumption rate (OCR) was measured by the mitochondrial stress test assay performed using the Seahorse XFe24 Bioanalyzer (Agilent Technologies, Santa Clara, CA, USA) as per the manufacturer’s instructions and as described previously [23]. Prior to the assay, the serum-free media was replaced with seahorse assay medium consisting of Seahorse XF DMEM base medium (Agilent, Santa Clara, CA, USA) supplemented with 10 mM glucose, 1 mM sodium pyruvate, and 2 mM glutamine at pH 7.4. A Seahorse XFe24 FluxPak sensor cartridge, containing solid-state metalloporphyrins complexes (MPCs) as the O_2_ sensor, was hydrated using Seahorse XF Calibrant Solution for 24 h prior to assaying. Titration of mitochondrial inhibitors and FCCP (Seahorse XF Cell Mito Stress Test kit) were performed across patients to determine the optimal concentrations that were subsequently used to test mitochondrial respiration. The optimal concentration of the inhibitors and FCCP in hASM cells remained consistent across patients and were as follows: 1 µM oligomycin (ATP uncoupler), 1.25 µM FCCP (accelerates electron transport chain), and 1 µM antimycin A (Complex III inhibitor) with 1 µM rotenone (Complex I inhibitor) [64,65,66,67,68,69]. OCR measurements obtained were normalized for total adherent cell count, obtained before and after assaying, or average mitochondrial volume density per cell [34,70]. The total number of adherent cells within each well was obtained by in situ cell counting prior to and after every mitochondrial stress test using 1 µg/mL Hoechst 33342 Solution (Invitrogen, Carlsbad, CA, USA; excitation/emission wavelength: 361/486 nm) and Cytation 5 Cell Lab Manager imaging system (BioTek, Winooski, VT, USA). Images of Hoechst-stained nuclei of hASM cells were captured as a single center image of the well covering an area of 7.41 mm^2^. Since the total area of the well covered by the adherent cells was 27.5 mm^2^, the total number of adherent cells in each well was estimated by extrapolating the counts obtained from the single center image by multiplying them by a factor of ~3.71 (ratio of the area of the well to the area of the image obtained). hASM cells from 6 patients were used for OCR measurement (*n* = 4 wells/patient).

### 4.10. Statistical Analysis

For each of the experiments, hASM cells were dissociated from bronchial tissue samples obtained from both female and male patients. Sex is an important biological variable, but the study was not powered to detect sex differences. The patient inclusion criteria were based on the phenotype of dissociated cells as confirmed by immunoreactivity to α-SMA. If less than 90% of the dissociated cells were not immunoreactive for α-SMA, the samples were discarded. Three out of nine patient samples were excluded based on these criteria, resulting in the inclusion of six patients. Differences across patients were considered a random variable in the statistical design. Sample size (n) represents the number of hASM cells or biological replicates analyzed per patient and the n for each experiment are provided in the figure legends. For SDH_max_ and mitochondrial volume density measurements, each dot represents data obtained from a single hASM cell and each color represents a single patient (6 patients—3 females and 3 males). For OCR measurements, each dot represents data obtained from one well (4 wells/patient) and each color represents a single patient (6 patients—3 females and 3 males). For measurements of SDH_max_ with varying number of optical slices, substrate dependence (in intact and permeabilized cells), and mitochondrial volume density (CellLight and MitoTracker), each dot represents a single cell from one patient. The number of cells per patient used was determined by a power analysis of primary outcome measures. The expected effect size was calculated with an a priori biologically relevant difference of 20% and equal variance, with sample size estimated using d = 1.4, α = 0.05, and β = 0.8. A Shapiro–Wilk test was used to confirm a normal distribution in the data. Statistical analysis was conducted using Prism 9 (GraphPad, San Diego, CA, USA), and a paired *t*-test was used to compare groups. The linearity of the SDH_max_ assay was assessed using the simple linear regression model. The linear relationship between SDH_max_, OCR, and mitochondrial volume density was assessed with Pearson’s correlation. Statistical significance (*) was concluded if *p* < 0.05. All data are represented as mean ± SEM or mean ± SD.

## 5. Conclusions

In conclusion, we present evidence that the quantitative histochemical technique for measuring SDH_max_ in hASM cells is reliable and reproducible. The SDH_max_ measurements were not comparable to respirometry-based OCR measurements, which is considered the “gold standard” for the measurement of mitochondrial function. However, we provide evidence that respirometry-based OCR measurements are more susceptible to variability than SDH_max_ measurements. We conclude that the quantitative histochemical technique to measure SDH_max_ can be used in other cell populations. Future studies targeting the SDH complex may unravel further information on the mechanism and specificity of the SDH assay.

## Figures and Tables

**Figure 1 ijms-24-11506-f001:**
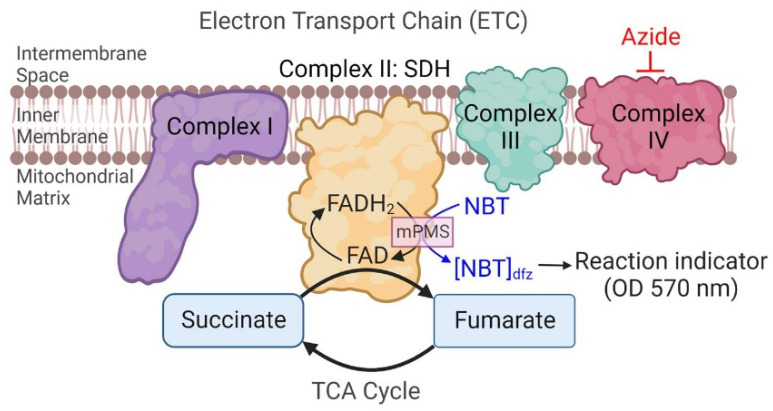
An overview of the quantitative histochemical technique for measurement of the maximum velocity of the succinate dehydrogenase reaction (SDH_max_). In the SDH reaction, the progressive reduction of nitroblue tetrazolium (NBT) to its diformazan (NBT_dfz_) is used as the reaction indicator. The reaction is performed in the presence of 1-methoxyphenazine methosulphate (mPMS), an exogenous electron carrier, and azide to inhibit cytochrome oxidase (Complex IV). The accumulation of NBT_dfz_ within a 3D region of interest in an hASM cell was measured every 15 s across a 10-min period as the change in OD at 570 nm. (Created with BioRender.com).

**Figure 2 ijms-24-11506-f002:**
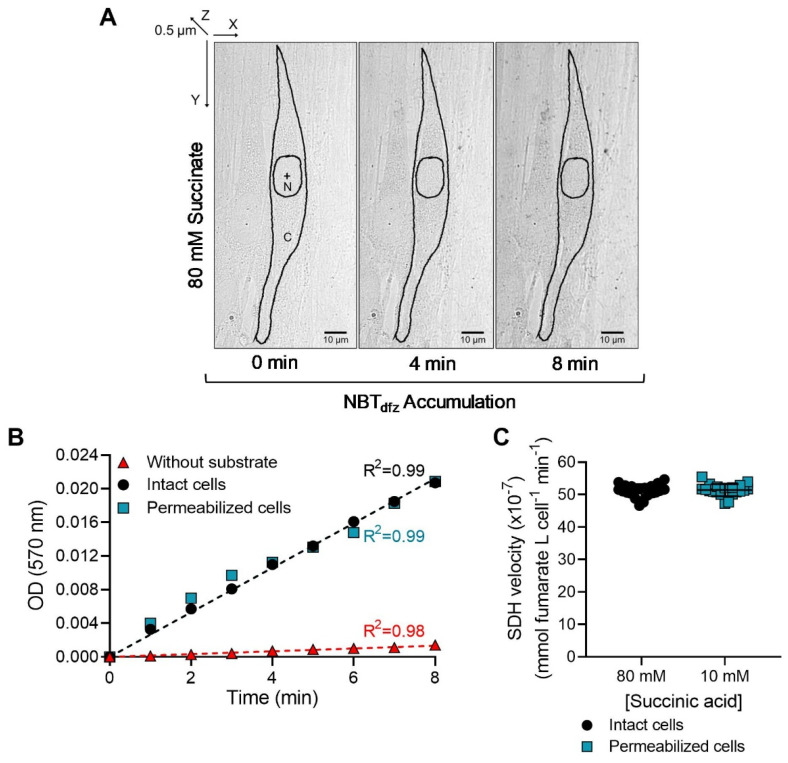
(**A**) Representative image shows the change in OD within a delineated intact hASM cell at 0, 4, and 8 min during the SDH reaction (Scale bar  =  10 μm). Individual hASM cells were delineated as the region of interest (ROI) for the measurement of change in OD while the nucleus was excluded from the ROI (N: Nucleus, C: Cytoplasm, +: Nuclear centroid computed using NIS-Elements software). (**B**) In intact and permeabilized hASM cells, the OD was measured every 15 s over a 10 min period, with and without substrate (succinate). In both intact and permeabilized hASM cells, the SDH reaction was found to be linear (R^2^ = 0.99) across an 8 min period in the presence of 80 mM and 10 mM substrate, respectively. The rate in the presence of substrate was significantly higher compared to the SDH reaction without substrate (*p* < 0.0001). Results were analyzed using a simple linear regression model. Circles represent SDH reaction performed in intact cells with 80 mM succinate, squares represent SDH reaction performed in permeabilized cells with 10 mM succinate, and triangles represent SDH reaction performed without succinate. (**C**) The SDH_max_ produced with 10 mM succinate in permeabilized cells was comparable to the SDH_max_ produced with 80 mM succinate in intact cells. Data are presented as mean ± SD in a scatter plot. The results represent SDH_max_ from one bronchial sample (patient). Circles represent SDH reaction performed in intact cells with 80 mM succinate and squares represent SDH reaction performed in permeabilized cells with 10 mM succinate. Statistical analyses in hASM cells were based on measurements from *n* = 20 hASM cells using a paired *t*-test.

**Figure 3 ijms-24-11506-f003:**
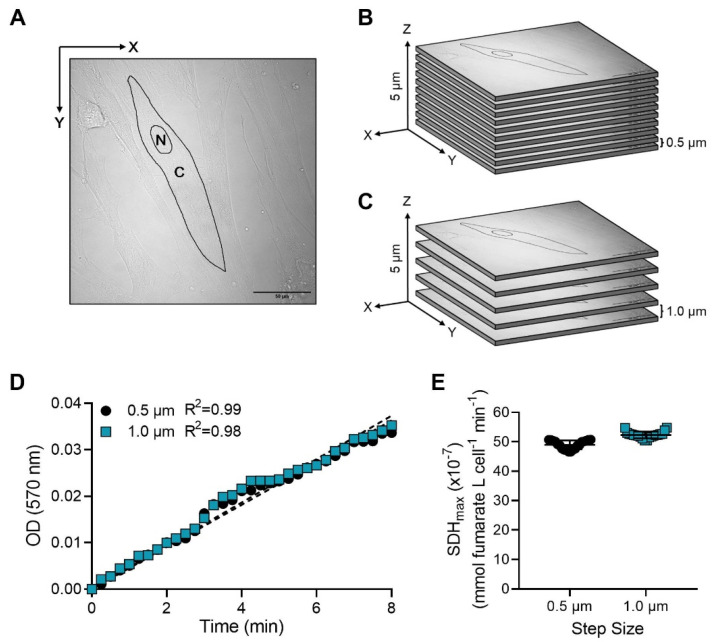
The reaction product (NBT_dfz_) is distributed within hASM cells. (**A**) Representative diagram showing a delineated ROI within an hASM cell centered at mid-nucleus (scale bar  =  50 μm). The nucleus was excluded from the ROI. (**B**,**C**) A Z-stack with a controlled optical slice thickness of 0.5 µm was centered at mid-nucleus and extended for 5 µm in the z-axis with a step size of 0.5 µm (10 optical slices) or 1.0 µm (5 optical slices), (N: nucleus, C: cytoplasm). (**D**) In intact hASM cells, the OD was measured every 15 s over a 10 min period. The SDH reaction was found to be linear across an 8 min period at 0.5 µm step size (R^2^ = 0.99) and 1.0 µm step size (R^2^ = 0.98). Results were analyzed using a simple linear regression model. Circles represent SDH reaction performed at 0.5 µm step size and squares represent SDH reaction performed at 1.0 µm step size. (**E**) The mean SDH_max_ from all optical slices obtained at a step size of 0.5 and 1.0 µm between optical slices was measured and the SDH_max_ within the total volume of the hASM cells across all patients was compared. No significant difference in SDH_max_ was observed between the two sampling parameters. Data are presented as mean ± SD in a scatter plot. Data represent results from one bronchial sample (patient), circles represent optical slices obtained at 0.5 µm step size, and squares represent optical slices obtained at 1.0 µm step size. Statistical analyses in hASM cells were based on measurements from *n* = 20 hASM cells per patient using a paired *t*-test.

**Figure 4 ijms-24-11506-f004:**
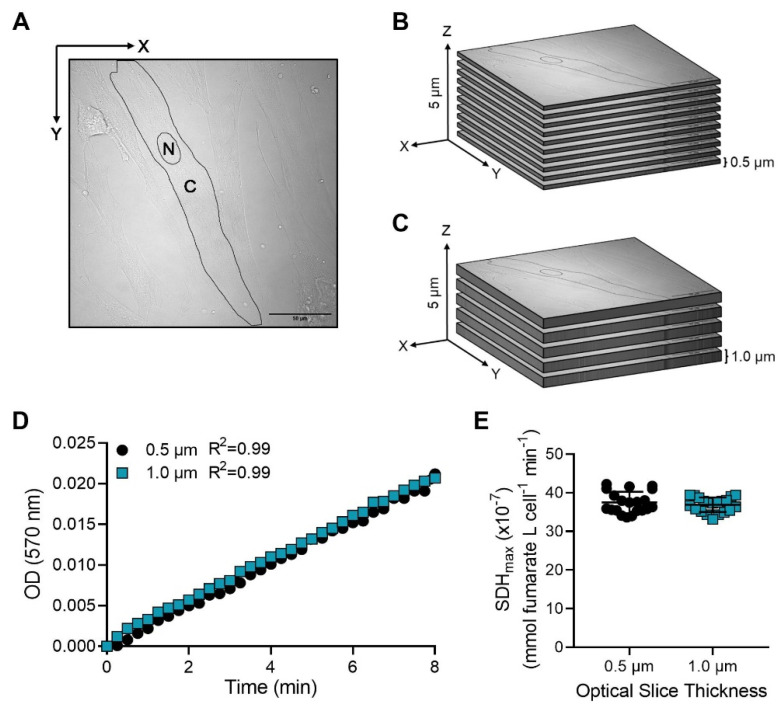
The NBT_dfz_ OD is proportional to pathlength. (**A**) Representative diagram showing a delineated ROI within an hASM cell centered at mid-nucleus in the XY plane (scale bar  =  50 μm). The nucleus was excluded from the ROI. (**B**,**C**) Z-stack with a controlled step size of 0.5 µm between optical slices was obtained with an optical slice thickness of 0.5 µm (10 optical slices) or 1.0 µm (5 optical slices), (N: nucleus, C: cytoplasm). (**D**) In intact hASM cells, the OD was measured every 15 s over a 10 min period. The SDH reaction was found to be linear (R^2^ = 0.99) across an 8 min period with both optical slice thicknesses, 0.5 µm and 1.0 µm. Results were analyzed using a simple linear regression model. Data represent results from one bronchial sample (patient), circles represent SDH reaction performed at an optical slice thickness of 0.5 µm, and squares represent SDH reaction performed at an optical slice thickness of 1.0 µm. (**E**) The mean SDH_max_ measured from all optical slices obtained with an optical slice thickness of 0.5 µm was comparable to the mean SDH_max_ measured with an optical slice thickness of 1.0 µm. Data are presented as mean ± SD in a scatter plot. Each dot represents results from an individual hASM cell, circles represent SDH reaction performed at an optical slice thickness of 0.5 µm, and squares represent SDH reaction performed at an optical slice thickness of 1.0 µm. Statistical analyses in hASM cells were based on measurements from *n* = 20 hASM cells from a single patient using a paired *t*-test.

**Figure 5 ijms-24-11506-f005:**
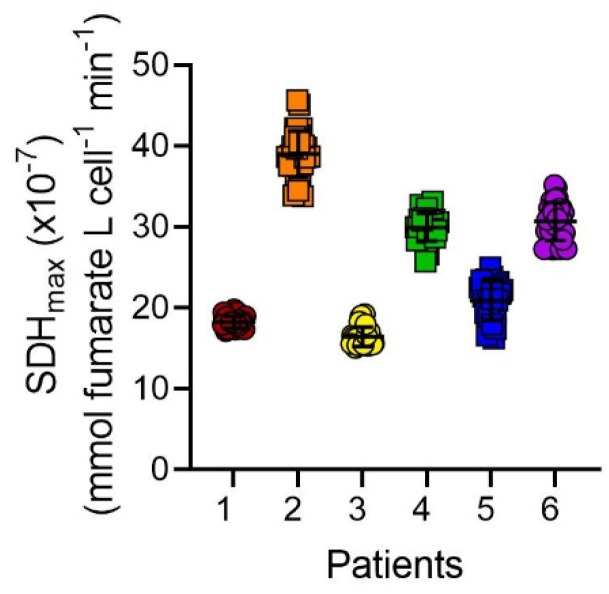
SDH_max_ is reproducible across different patients. The range of SDH_max_ within hASM cells was measured across different patient samples. SDH_max_ measurements averaged 25.86 × 10^−7^ ± 1.88 mmol fumarate L cell^−1^ min^−1^ across patients while showing low variability within each patient when compared to that across patients. Data are presented as mean ± SD in a scatter plot. Each color represents results from one bronchial sample (patients). Circles represent hASM cells dissociated from male patients and squares represent hASM cells dissociated from female patients. For each patient, SDH_max_ was measured in *n* = 30 hASM cells per patient from 6 bronchial samples (patients).

**Figure 6 ijms-24-11506-f006:**
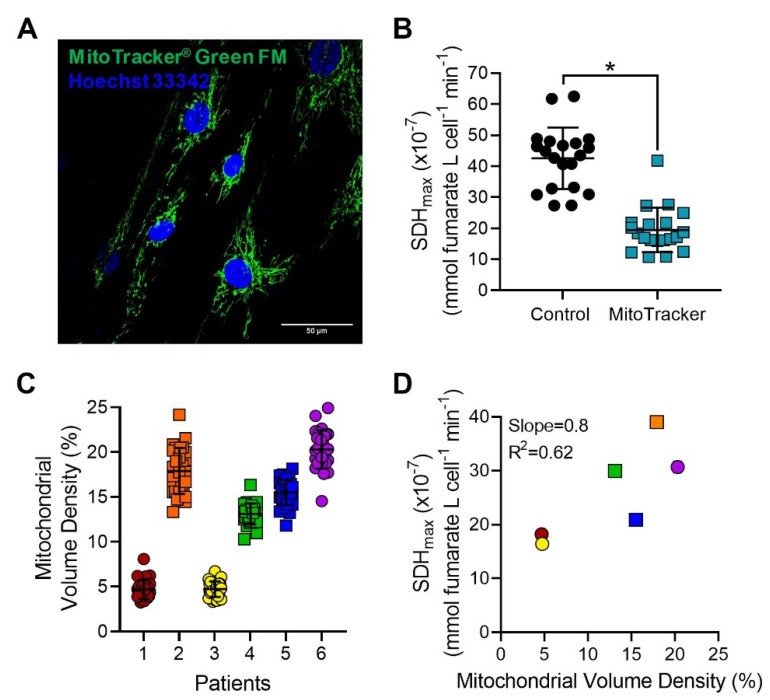
MitoTracker labeling decreases SDH_max_ measurement. (**A**) Representative maximum intensity Z-projection image of hASM cells loaded with MitoTracker^®^ Green FM to visualize mitochondria. Multiple hASM cells were visualized within a single microscopic field and were used for mitochondrial volume density measurements (scale bar  =  50 μm). (**B**) In delineated hASM cells, SDH_max_ was measured after labeling with MitoTracker^®^ Green FM. SDH_max_ in hASM cells loaded with MitoTracker was decreased compared to unlabeled hASM cells (* *p* < 0.0001). Each dot represents results from an individual hASM cell, circles represent SDH_max_ measurements from unlabeled (control) cells, and squares represent SDH_max_ measurements from MitoTracker labeled cells. Data are presented as mean ± SD in a scatter plot. Statistical analyses in hASM cells were based on measurements from *n* = 20 hASM cells per group from a single patient using paired-t test. (**C**) Mitochondrial volume density was measured using Z-stack fluorescent images of hASM cells and 3D reconstruction using ImageJ. The mitochondrial volume densities varied within patients with an average density of ∼14% across patients. Data are presented as mean ± SD in a scatter plot. Each color represents results from one bronchial sample (patient). Circles represent hASM cells dissociated from male patients and squares represent hASM cells dissociated from female patients. For each patient, mitochondrial volume density was measured in *n* = 30 hASM cells per patient from 6 bronchial samples (patients). (**D**) SDH_max_ showed a positive linear relationship with mitochondrial volume density but was not significant. Scatterplot shows the relationship between SDH_max_ (*y*-axis) and mean mitochondrial volume density (*x*-axis) across all hASM patients (slope = 0.8, R^2^ = 0.62, *p* > 0.05) determined by Pearson’s correlation. Circles represent hASM cells dissociated from male patients, and squares represent hASM cells dissociated from female patients.

**Figure 7 ijms-24-11506-f007:**
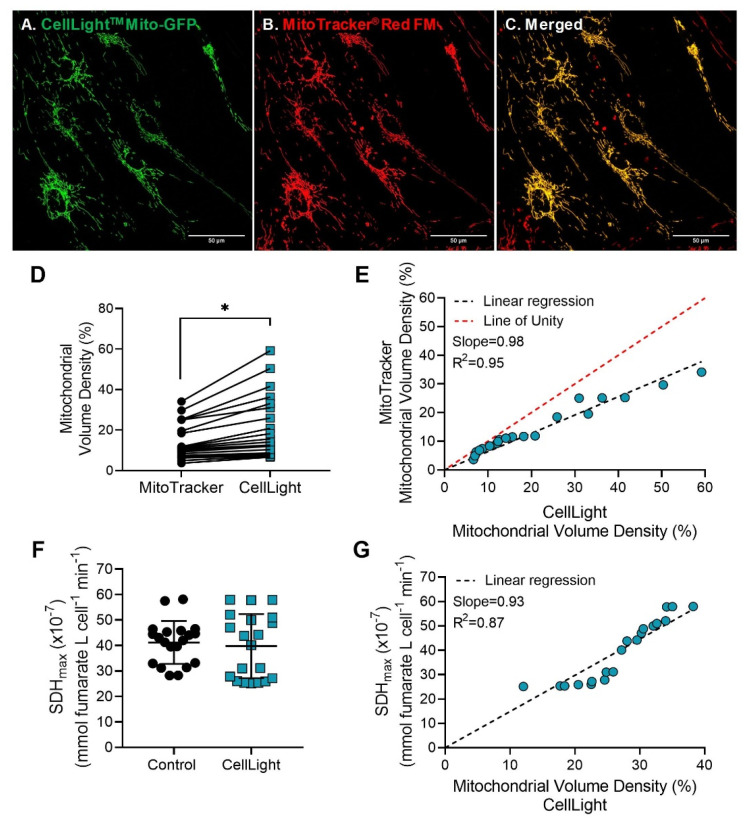
SDH_max_ is dependent on mitochondrial volume density. (**A**,**B**) Representative maximum intensity Z-projection image of hASM cells transduced with CellLight^™^ Mitochondria-GFP and labeled with MitoTracker^®^ Red FM to visualize mitochondria (scale bar  =  50 μm). (**C**) Overlay of the images of the two fluorescence labels confirmed that the pattern of CellLight labeling was restricted to mitochondria, similar to MitoTracker. (**D**) The mitochondrial volume density was measured using the 3D reconstruction of Z-stack fluorescent images of hASM cells, loaded with CellLight and MitoTracker, using ImageJ. Mitochondrial volume density measured in CellLight labeled hASM cells was significantly higher compared to that measured in MitoTracker labeled hASM cells (* *p* < 0.001). Each dot represents mitochondrial volume density of an individual hASM cell from a single patient with lines showing the change in mitochondrial volume density measured in the same hASM cell labeled with CellLight. Circles represent measurements from MitoTracker labeled hASM cells and squares represent measurements from CellLight transduced hASM cells. (**E**) Scatterplot shows the positive linear relationship between the mitochondrial volume density measured with CellLight (*y*-axis) and MitoTracker (*x*-axis) across cells (slope = 0.98, R^2^ = 0.95, *p* < 0.0001) determined by Pearson’s correlation. Statistical analyses in hASM cells were based on measurements from *n* = 20 hASM cells per group from a single patient. (**F**) SDH_max_ was measured in individual hASM cells after CellLight labeling. SDH_max_ in CellLight labeled hASM cells was unchanged compared to untransduced (control) cells. Each dot represents results from an individual hASM cell, circles represent SDH_max_ measurements from untransduced (control) cells, and squares represent SDH_max_ measurements from CellLight transduced cells. Data are presented as mean ± SD. Statistical analyses in hASM cells were based on measurements from *n* = 20 hASM cells per group from a single patient using a paired-*t* test. (**G**) SDH_max_ showed a significant positive linear relationship with mitochondrial volume density measured by CellLight labeling. Scatterplot shows the relationship between SDH_max_ (*y*-axis) and mitochondrial volume density (*x*-axis) across cells (slope = 0.93, R^2^ = 0.87, *p <* 0.0001) determined by Pearson’s correlation.

**Figure 8 ijms-24-11506-f008:**
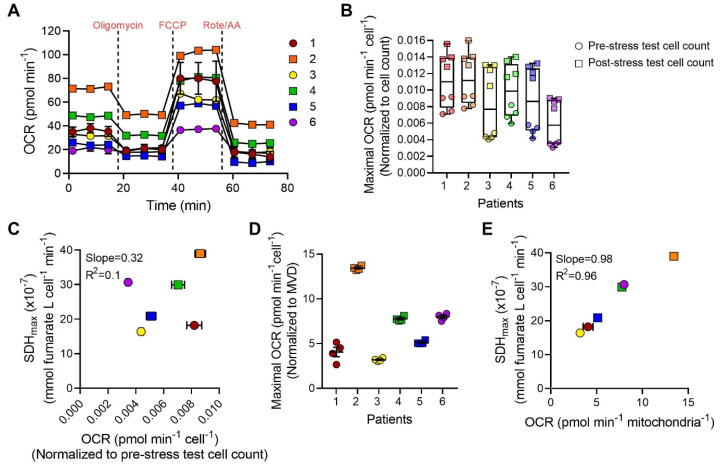
O_2_ consumption rate (OCR) normalized to mitochondrial volume density correlates with SDH_max_ in hASM cells. (**A**) The raw XF traces of the mitochondrial stress test were performed with the sequential use of 1 µM oligomycin (ATP uncoupler), 1.25 µM FCCP (proton ionophore), and 1 µM antimycin A (Complex III inhibitor) with 1 µM rotenone (Complex I inhibitor). Data are presented as mean ± SEM. Each color represents results from one bronchial sample (patient). Circles represent hASM cells dissociated from male patients and squares represent hASM cells dissociated from female patients. For each patient, OCR was measured in *n* = 4 wells per patient, from 6 bronchial samples (patients). (**B**) The maximum OCR in hASM cells was normalized to the total cell count obtained before and after assaying. Maximum OCR values across patients ranged between 0.0061 ± 0.0006 and 0.012 ± 0.001 pmol min^−1^ cell^−1^ when normalized to cell count before and after assay, respectively. Data are presented as box-whisker plots showing the median and minimum to maximum distribution of maximum OCR from all six patients. Each color represents results from one bronchial sample (patient), circles represent maximum OCR normalized to cell count obtained before stress test, and squares represent maximum OCR normalized to cell count obtained after stress test. (**C**) SDH_max_ was not correlated with maximum OCR per cell. Scatterplot shows the relationship between SDH_max_ (y-axis) and mean OCR per cell (x-axis) across all hASM patients (slope = 0.32, R^2^ = 0.10, *p* > 0.05) determined by Pearson’s correlation. Circles represent hASM cells dissociated from male patients and squares represent hASM cells dissociated from female patients. (**D**) Maximum OCR in hASM cells was normalized for mitochondrial volume density (MVD). Data are presented as mean ± SEM in a scatter plot. Each color represents results from one bronchial sample (patient). Circles represent hASM cells dissociated from male patients and squares represent hASM cells dissociated from female patients. (**E**) Scatterplot shows the positive linear relationships between SDH_max_ (y-axis) and mean OCR per mitochondrion (x-axis) across all hASM patients (slope = 0.98, R^2^ = 0.96, *p* < 0.01) determined by Pearson’s correlation. Circles represent hASM cells dissociated from male patients and squares represent hASM cells dissociated from female patients.

**Figure 9 ijms-24-11506-f009:**
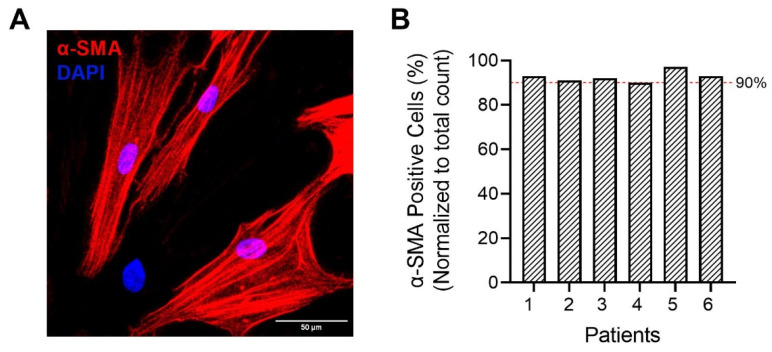
Confirmation of hASM phenotype. (**A**) Representative maximum intensity Z-projection image of dissociated cells. The phenotype of dissociated human airway smooth muscle (hASM) cells was assessed based on immunoreactivity to α-smooth muscle actin (α-SMA) expression. α-SMA immunoreactive hASM cells were larger (scale bar  =  50 μm). (**B**) Column bar graph represents the percentage of hASM cells determined as the fraction of α-SMA expressing cells relative to the total dissociated cells (determined from DAPI) from 6 patient samples (*n* = 120 hASM cells per patient). Note that ~95% of total dissociated cells were immunoreactive to α-SMA.

**Figure 10 ijms-24-11506-f010:**
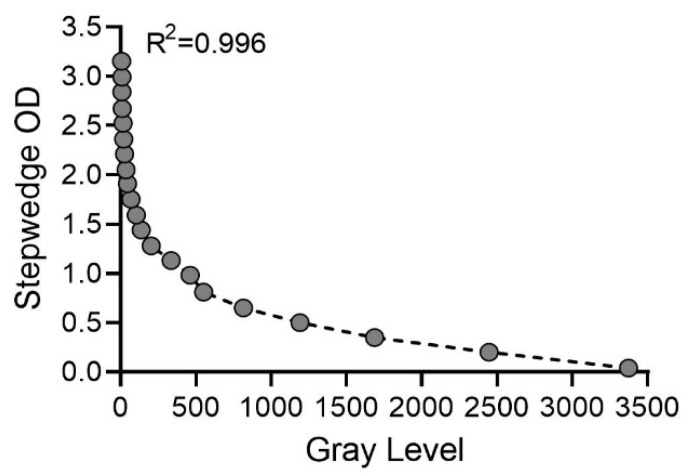
Calibration of gray level (GL) to known optical density (OD). The measured gray level (GL) of the microscope was calibrated to known optical density (OD) units using a photographic density stepwedge tablet. The dynamic range during image acquisition was adjusted to take advantage of the 4096 gray levels in the 12-bit system. The GL of the acquisition system was measured and plotted to the known OD units provided in the stepwedge tablet (0.04–2.20 OD units in increments of 0.15 OD). The data obtained were analyzed by a four-parameter general curve fit function to derive an equation (Equation (1)), which was used to transform measured GL values to OD in unknown samples.

**Table 1 ijms-24-11506-t001:** Reproducibility of measurement of the velocity of SDH reaction across patients.

Donor No.	SDH_max_ (×10^−7^) mmol Fumarate L cell^−1^ min^−1^			
	Intact Cells		Permeabilized Cells	
	Mean ± SD	CV%	Mean ± SD	CV%
1	18.22 ± 0.74	4	18.56 ± 0.88	5
2	16.42 ± 2.87	7	16.36 ± 0.53	3
3	39.03 ± 1.23	7	40.50 ± 1.46	4
4	29.96 ± 1.69	5	29.58 ± 1.68	6
5	20.89 ± 2.40	6	21.66 ± 1.28	6
6	30.69 ± 2.35	7	31.86 ± 1.08	4
Mean	25.86 ± 1.88	6	26.42 ± 1.15	4

**Table 2 ijms-24-11506-t002:** Mitochondrial volume density measurements across patients.

Donor No.	Mitochondrial Volume Density	
	Mean% ± SD	CV%
1	4.70% ± 1.05	22
2	17.91% ± 2.57	14
3	4.74% ± 0.86	18
4	13.44% ± 1.17	8
5	15.51% ± 1.42	9
6	20.30% ± 2.13	10
Mean	12.70% ± 1.53	14

**Table 3 ijms-24-11506-t003:** OCR measurements across patients.

Donor No.	Oxygen Consumption Rate pmol min^−1^					
	Normalized to Cell Count Pre-Stress Test (×10^−3^)		Normalized to Cell Count Post-Stress Test (×10^−3^)		Normalized to Mitochondrial Volume Density	
	Mean ± SD	CV%	Mean ± SD	CV%	Mean ± SD	CV%
1	8.20 ± 1.07	13	14.03 ± 1.16	8	4.06 ± 1.06	26
2	8.63 ± 0.76	9	14.20 ± 1.27	9	13.43 ± 0.24	3
3	4.38 ± 0.33	7	12.24 ± 1.19	10	3.20 ± 0.17	6
4	7.04 ± 0.93	13	12.77 ± 1.18	9	7.56 ± 0.26	4
5	5.11 ± 0.70	14	12.40 ± 0.80	6	5.12 ± 0.19	4
6	3.44 ± 0.28	8	8.58 ± 0.55	6	8.00 ± 0.36	5
Mean	6.14 ± 0.68	11	12.37 ± 1.02	8	6.93 ± 0.38	8

## Data Availability

The raw data supporting the conclusions of this article will be made available by the authors, without undue reservation.

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
