# Peer review of "Cell-Based Measurement of Mitochondrial Function in Human Airway Smooth Muscle Cells"

_ijms, 2023, doi:10.3390/ijms241411506_

Round 1

Reviewer 1 Report

The authors assessed single cell measurement of mitochondrial function in human airway smooth muscle cells using quantitative histochemical technique to measure the activity of succinate dehydrogenase. Although this technique is rigorous and reproducible there are major concerns with terminologies that the authors should consider changing to accurately represent their findings.

1.       The authors should consider changing the title from “single cell” to “cell-type” measurement. Authors isolated cells from airway smooth muscle cells and then measured mitochondrial function. Even though in their confocal measurement they are looking at one cell at time, it is not single cell measurement but a cell-type.

2.       It is difficult to assess the validity of this new technique that measure activity of SDH because the authors did not show how their new method compared with the existing method for assessing the activity of SDH in intact cell.

3.       Authors in their discussion repeat methods of data collection e.g in lines 350-370 and in other parts of the discussion.

Author Response

Response to Reviewer 1 Comments

Comment 1: The authors assessed single cell measurement of mitochondrial function in human airway smooth muscle cells using quantitative histochemical technique to measure the activity of succinate dehydrogenase. Although this technique is rigorous and reproducible there are major concerns with terminologies that the authors should consider changing to accurately represent their findings.

Response 1: We thank the reviewer for the succinct summary of our study.

Comment 2: The authors should consider changing the title from “single cell” to “cell-type” measurement. Authors isolated cells from airway smooth muscle cells and then measured mitochondrial function. Even though in their confocal measurement they are looking at one cell at time, it is not single cell measurement but a cell-type.

Response 2: Respectfully, we are confused by the reviewer’s comment. As noted by the reviewer, confocal measurements of SDH reaction rate were made “one cell at a time”. We don’t fully understand why one cell at a time “is not single cell measurements”. While the measurements in the current study were restricted to hASM cells, we do not imply that this method is specific to hASM cells only, i.e., it is not “cell-type” restricted. The goal of using confocal microscopy to measure SDHmax was to provide an assessment of mitochondrial function within individual cells regardless of cell type, hence the use of the term “single cell” measurement. However, in response to the reviewer’s comment, we have revised the title to “Cell-Based Measurements of Mitochondrial Function in Human Airway Smooth Muscle Cells”.

Comment 3: It is difficult to assess the validity of this new technique that measure activity of SDH because the authors did not show how their new method compared with the existing method for assessing the activity of SDH in intact cell.

Response 3: There is no “existing method for assessing the activity of SDH in intact cell”. There are existing biochemical methods for assessing the velocity of the SDH reaction in cell and tissue lysates or isolated mitochondria. Importantly, these biochemical methods work with the same principle as our confocal imaging method. We now emphasize that the underlying principles of our cell-based measurements of SDHmax are comparable to those in lysate-based biochemical measurements.

Unlike these existing methods, this study employs the use of intact live cells cultured in vitro and provides us with a time-based kinetic measurement of the maximum SDH velocity in individual cells and avoids any possible artifacts that could be introduced. As a more accurate comparison for functional measurement in intact live cells cultured in vitro, we used the mitochondrial stress test (respirometry) to measure O2 consumption rate (OCR). Where from the data obtained, we rigorously assessed the correlation between OCR and SDHmax.

Comment 4: Authors in their discussion repeat methods of data collection e.g., in lines 350-370 and in other parts of the discussion.

Response 4: In response to the reviewer’s comment, we have removed details of our methods from the Discussion section and included these (if not already stated) in the Methods section.

Reviewer 2 Report

The study by Bhat et al. intended to validate a novel optical sectioning-based technique to measure SDHmax in individual human airway smooth muscle cells (hASMC) by assessing the progressive SDH-dependent precipitation of NBTdfz. SDHmax varied with mitochondrial volume density across hASMC and correlated with the mitochondrial volume density.

General comments

The study is well-conducted and technically sound. The authors introduce an innovative and reproducible optical method to measure SDHmax in mitochondria of living cells by the use of confocal microscopy.

Author Response

Response to Reviewer 2 Comments

Comment 1: The study by Bhat et al. intended to validate a novel optical sectioning-based technique to measure SDHmax in individual human airway smooth muscle cells (hASMC) by assessing the progressive SDH-dependent precipitation of NBTdfz. SDHmax varied with mitochondrial volume density across hASMC and correlated with the mitochondrial volume density.

Response 1: We thank the reviewer for the succinct summary of our study.

Comment 2: The study is well-conducted and technically sound. The authors introduce an innovative and reproducible optical method to measure SDHmax in mitochondria of living cells by the use of confocal microscopy.

Response 2: We thank the reviewer for the very positive comments.

Round 2

Reviewer 1 Report

The authors have addressed all my major concerns on the manuscript.